



# Training a supermodel with noisy and sparse observations: a case study with CPT and the synch rule on SPEEDO - v.1

Francine Schevenhoven[1,2] and Alberto Carrassi[3,4]

[1]Geophysical Institute, University of Bergen, Bergen, Norway
[2]Bjerknes Centre for Climate Research, Bergen, Norway
[3]Dept. of Meteorology and NCEO, University of Reading, United Kingdom
[4]Mathematical Institute, Utrecht University, Utrecht, the Netherlands

**Correspondence:** Francine Schevenhoven (francine.schevenhoven@uib.no)

**Abstract.** In alternative to using the standard multi-model ensemble (MME) approach to combine the output of different models to improve prediction skill, models can also be combined dynamically to form a so-called supermodel. The supermodel approach allows for a quicker correction of the model errors. In this study we connect different versions of SPEEDO, a global atmosphere-ocean-land model of intermediate complexity, into a supermodel. We focus on a weighted supermodel, in which the

supermodel state is a weighted superposition of different imperfect model states. The estimation, "the training", of the optimal weights of this combination is a critical aspect in the construction of a supermodel. In our previous works two algorithms were developed: (i) cross pollination in time (CPT-based technique), and, (ii) a synchronization based learning rule (synch rule). Those algorithms have been so far applied under the assumption of complete and noise-free observations. Here we go beyond and consider the more realistic case of noisy data that do not cover the full system's state and are not taken at each

model's computational time step. We revise the training methods to cope with this observational scenario, while still being able to estimate accurate weights. In the synch rule an additional term is introduced to maintain physical balances, while in CPT nudging terms are added to let the models stay closer to the observations during training. Furthermore, we propose a novel formulation of the CPT method allowing for the weights to be negative. This makes it possible for CPT to deal with cases in which the individual model biases have the same sign, a situation that hampers constructing a skilful weighted supermodel

based on positive weights. With these developments, both CPT and the synch rule have been made suitable to train a supermodel consisting of state-of-the-art weather or climate models.

## 1 Introduction

Climate models are continuously improving over time. This is made evident by the succession of the Coupled Model Inter-comparison Project (CMIP), which is currently at its sixth stage (Eyring et al., 2016). The CMIP models are used by the

Intergovernmental Panel on Climate Change (IPCC) for its assessment reports. The model complexity is increasing and more processes can be resolved due to increased spatial and temporal resolutions. Nevertheless, the real climate system is too complex (Ghil and Lucarini, 2020) for any numerical model so that models will remain inevitably imperfect (Palmer and Stevens, 2019).





Given a set of imperfect models, one can combine them so that their combination has higher forecast skill than each in-
dividual model independently. A common approach is to use the multi-model-ensemble (MME) (Hagedorn et al., 2005). In
the MME the individual model ensembles are constructed based on different initial conditions but propagated forward in time
using the same model. After the integration, the ensembles from different models are combined. The MME is a very powerful
and useful approach to achieve better statistics such as the mean; this is because errors tend to cancel each other (Hagedorn
et al., 2005). Generally, the models are equally weighted in a MME, as is the case for, *e.g.* the CMIP runs in the IPCC re-
ports. Another possibility is to calculate a so-called "super-ensemble" (Krishnamurti et al., 2016), where the model weights are
trained on the basis of historical observations, *e.g.* in Hagedorn et al. (2005) and Doblas-Reyes et al. (2005). MME is inherently
a statistical method that does not take into account possible changes in the model regimes. A caveat is obviously that weights
that are optimal for model behavior in the past do not necessarily convert into optimal weights for the future. To cope with this,
a "dynamical" on-the-fly approach to combine models is desirable.

Along this line, in the supermodel approach, models are combined during the simulation by sharing their own tendencies,
and not just their outputs as with the MME. This amounts to creating a new virtual model, the supermodel, that can potentially
have better physical behaviour than the individual models. By combining the models dynamically into a supermodel, model
errors can be reduced at an earlier stage, potentially mitigating error propagation. This is particularly helpful since the climate
system is not linear, which causes initial errors to spread over different variables and regions. The simulated climate statistics of
the supermodel are therefore expected to be superior to that from the combination of biased models. The supermodel not only
improves the statistics of simulated climate, as in the MME, it can also give an improved model trajectory if the models are
well enough synchronized. This could be essential in order to predict a specific sequence of weather or climate events. Given
that the individual model trajectories in MME are "free" to evolve according to each of the model dynamics, their averaging
may result in an overall cancellation of the individual variabilities.

The supermodeling approach was originally developed using low-dimensional dynamical systems (van den Berge et al.,
2011; Mirchev et al., 2012) and subsequently applied to a global quasi-geostrophic atmospheric model (Schevenhoven and Sel-
ten, 2017; Wiegerinck and Selten, 2017) and to a coupled atmosphere-ocean-land model of intermediate complexity SPEEDO
(Selten et al., 2017; Schevenhoven et al., 2019). A partial supermodel implementation using state-of-the-art coupled ocean-
atmosphere models and using real-world observations was presented in Shen et al. (2016). A crucial step in supermodeling
is the training of the weights based on data. The first supermodel training schemes were based on the minimization of a cost
function (van den Berge et al., 2011; Shen et al., 2016), an approach with high computational cost, relying on a large number
of long model runs. Schevenhoven and Selten (2017) developed a computationally efficient training scheme based on Cross
Pollination in Time (CPT), a concept originally introduced by Smith (2001). In CPT, the models in a multi-model ensemble
exchange states during the simulation. As a consequence, the CPT trajectory tends to explore a larger area of the phase space
than the individual models, thus enhancing the chance to pass in the vicinity of an observation. Another efficient training
method, referred to as the "synch rule", was introduced by Selten et al. (2017). The method, originally developed by Duane
et al. (2007) for parameter estimation, is based on the synchronization theory of different systems.





The SPEEDO experiments in Selten et al. (2017) and Schevenhoven et al. (2019) were applied in a noise-free observation framework. The "historical observations", used to train the supermodel, were available at every model time step. In this paper, we make a step forward towards applying CPT and the synch rule in state-of-the-art models and real-world observations. Real-world observations are not perfect and are not continuously available in time. We adapt the training methods, again in the context of SPEEDO, in order to produce accurate weights, in the context of sparse observations affected by Gaussian distributed noise.

The paper is structured as follows. Section 2 briefly describes the SPEEDO model, and redefines the definition of the weighted supermodel in the context of sparse in time observations. Section 3 describes the training schemes CPT and the synch rule as used in Schevenhoven et al. (2019), and introduces adaptations to the methods to cope with sparse and noisy observations. In Schevenhoven et al. (2019), the synch rule was able to produce negative weights, and this seemed very beneficial in case models share biases that cannot compensate for each other. In this paper, we also explore the possibility of negative weights for CPT. Section 4 presents this possibility, together with the results of the adaptations to CPT and the synch rule in order to make the methods suitable for training on the basis of sparse and noisy observations. We conclude in Sect. 5 with a comparison of both training methods and an outlook to their application in state-of-the-art models.

## 2 Weighted supermodel

This section recalls the general structure of a weighted supermodel as defined in Schevenhoven et al. (2019), and summarizes the supermodel structure used with the coupled atmosphere-ocean-land model SPEEDO (Severijns and Hazeleger, 2010); full details can be found in Schevenhoven et al. (2019). We shall then describe how the supermodel formulations are modified to handle time-sparse noisy data.

In Schevenhoven et al. (2019) the weighted supermodel was defined by combining the tendencies of the individual models. In the case of two imperfect models with parametric error, the weighted supermodel reads:

$$\dot{\boldsymbol{x}}_1 = \boldsymbol{f}(\boldsymbol{x}_s, \boldsymbol{p}_1) \tag{1a}$$

$$\dot{\boldsymbol{x}}_2 = \boldsymbol{f}(\boldsymbol{x}_s, \boldsymbol{p}_2) \tag{1b}$$

$$\dot{\boldsymbol{x}}_s = \mathbf{W}_1 \dot{\boldsymbol{x}}_1 + \mathbf{W}_2 \dot{\boldsymbol{x}}_2, \tag{1c}$$

where $\boldsymbol{x}_s \in \mathbb{R}^n$ represents the supermodel state vector, $\boldsymbol{f}$ the nonlinear evolution function depending on the state $\boldsymbol{x}$ and on a number of adjustable parameters $\boldsymbol{p}_{1,2} \in \mathbb{R}^m$, and the diagonal matrices $\mathbf{W}_{1,2} = \mathrm{diag}(\boldsymbol{w}_{1,2})$ with $\boldsymbol{w}_{1,2} \in \mathbb{R}^n$ denote the weights. Training a weighted supermodel implies training the weights $\boldsymbol{w}$. In Schevenhoven et al. (2019), the tendencies were combined at each model's computational time step, $\delta t$, that was assumed to be the same among the imperfect models. This choice implied a substantial computational cost (since the models are combined at each time step), but it also supposes that observations were available at each time step, a condition rarely, if not never, encountered in real applications. Constructing a supermodel for real model and observational scenarios requires relaxing this assumption.





This leads us to redefine a weighted supermodel by combining individual models at every arbitrary $\Delta T > \delta t$, such that:

$$\dot{\boldsymbol{x}}_1 = \delta_{\mathrm{mod}(t,\Delta T)} \boldsymbol{f}(\boldsymbol{x}_s, \boldsymbol{p}_1) + (1 - \delta_{\mathrm{mod}(t,\Delta T)}) \boldsymbol{f}(\boldsymbol{x}_1, \boldsymbol{p}_1) \tag{2a}$$

$$\dot{\boldsymbol{x}}_2 = \delta_{\mathrm{mod}(t,\Delta T)} \boldsymbol{f}(\boldsymbol{x}_s, \boldsymbol{p}_2) + (1 - \delta_{\mathrm{mod}(t,\Delta T)}) \boldsymbol{f}(\boldsymbol{x}_2, \boldsymbol{p}_2) \tag{2b}$$

$$\boldsymbol{x}_s = \mathbf{W}_1 \boldsymbol{x}_1 + \mathbf{W}_2 \boldsymbol{x}_2 \quad \text{if} \quad \delta_{\mathrm{mod}(t,\Delta T)} = 1, \tag{2c}$$

where, the Kronecker $\delta$ function takes the value 1 when $mod(t, \Delta T) = 0$, and zero otherwise. In the latter case no super-
model state is defined. Note that, in contrast to the original formulation of the weighted supermodel given in Eq. (1), here
the individual model states are combined instead of their tendencies. In fact, combining the model tendencies every $\Delta T > \delta t$
does not result in a synchronized supermodel state, thus leading to a supermodel with poor forecast skill. Weighting the states
ensures a synchronized supermodel state every $\Delta T$.

## 2.1   SPEEDO model

The coupled model SPEEDO consists of an atmospheric component (SPEEDY), that exchanges information with a land (LBM)
and an ocean-sea-ice component (CLIO). Detailed descriptions of SPEEDO can be found in Severijns and Hazeleger (2010)
and Selten et al. (2017). SPEEDY describes the evolution of the horizontal wind components $U$ (east-west) and $V$ (north-south),
temperature $T$ and specific humidity $q$ at eight vertical levels plus the surface pressure $p_s$. The horizontal grid resolution has a
spacing of $3.75°$ ($48 \times 96$ grid cells). SPEEDY exchanges moisture and heat with the land model, LBM, which uses three soil
layers and up to two snow layers to close the hydrological cycle over land. The horizontal discretization of the LBM is the same
as for SPEEDY. Moreover, SPEEDY exchanges heat, water, and momentum with the ocean model, CLIO. CLIO describes the
evolution of ocean currents, temperature and salinity on a computational grid with $3°$ horizontal resolution and 20 unevenly
spaced layers in the vertical. A three-layer thermodynamic-dynamic sea-ice model describes the evolution of sea ice.

The SPEEDO equations can be formally and compactly written as

$$\dot{\boldsymbol{a}} = \boldsymbol{f}^{\mathrm{a}}(\boldsymbol{a}; \boldsymbol{p}^{\mathrm{a}}) + \boldsymbol{g}^{\mathrm{a}}(\boldsymbol{e}^{\mathrm{h}}, \boldsymbol{e}^{\mathrm{w}}, \boldsymbol{e}^{\mathrm{m}}) \tag{3a}$$

$$\dot{\boldsymbol{o}} = \boldsymbol{f}^{\mathrm{o}}(\boldsymbol{o}; \boldsymbol{p}^{\mathrm{o}}) + \boldsymbol{g}^{\mathrm{o}}(\mathcal{P}^{\mathrm{o}} \boldsymbol{e}^{\mathrm{h}}, \mathcal{P}^{\mathrm{o}} \boldsymbol{e}^{\mathrm{w}}, \mathcal{P}^{\mathrm{o}} \boldsymbol{e}^{\mathrm{m}}, \mathcal{P}^{\mathrm{o}} \boldsymbol{r}) \tag{3b}$$

$$\dot{\boldsymbol{l}} = \boldsymbol{f}^{\mathrm{l}}(\boldsymbol{l}; \boldsymbol{p}^{\mathrm{l}}) + \boldsymbol{g}^{\mathrm{l}}(\mathcal{P}^{\mathrm{l}} \boldsymbol{e}^{\mathrm{h}}, \mathcal{P}^{\mathrm{l}} \boldsymbol{e}^{\mathrm{w}}, \boldsymbol{r}), \tag{3c}$$

where $\boldsymbol{a}$ stands for atmosphere, $\boldsymbol{o}$ for ocean/sea-ice and $\boldsymbol{l}$ for land; $\boldsymbol{e}^{\mathrm{h}}$ represents the heat exchange between atmosphere and
surface, $\boldsymbol{e}^{\mathrm{w}}$ the water exchange, $\boldsymbol{e}^{\mathrm{m}}$ the momentum exchange and $\boldsymbol{r}$ the river outflow describing the stream of water from land
to ocean. The exchange vectors depend on the state of the atmosphere and the surface, but this dependency is not made explicit
in Eq. (3) to simplify the notation. The projection operators $\mathcal{P}$ represent the conservative regridding operations between the
computational grids of the different model components. The nonlinear functions $\boldsymbol{f}$ represent the cumulative contribution of
the modeled physical processes to the change in the state vectors, and depend on the values of the parameter vectors $\boldsymbol{p}$. The





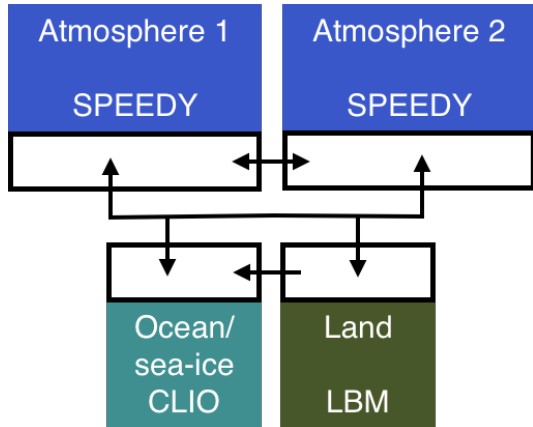

**Figure 1.** Schematic representation of the SPEEDO climate supermodel based on two imperfect atmospheric models. The two atmospheric models exchange water, heat and momentum with the perfect ocean and land model. The ocean and land model send their state information to both atmospheric models. The atmospheric models exchange state information in order to combine their states.

nonlinear functions $\boldsymbol{g}$ describe how the exchange of heat, water and momentum between atmosphere, ocean and land affects the change of the state vectors.

### 2.1.1 Weighted supermodel based on SPEEDO

A supermodel based on SPEEDO is formed by combining imperfect atmosphere components SPEEDY through a weighted superposition of the states of the imperfect models. All imperfect atmospheres are each coupled to the same ocean and land model. Figure 1 provides a schematic representation of the supermodel we construct.

All the atmospheric components of the individual imperfect models receive the same state information from ocean and land. Nevertheless, each atmosphere calculates its own water, heat and momentum exchange. Conversely, the ocean and land components receive the multi-model weighted average of the atmospheric states. This supermodel construction is inspired by the interactive ensemble approach originally devised by Kirtman and Shukla (2002).

We can now write the SPEEDO weighted supermodel equations as

$$\dot{\boldsymbol{a}}_{i} = [\delta_{\mathrm{mod}(t,\Delta T)} \boldsymbol{f}^{a}(\boldsymbol{a}_{s}; \boldsymbol{p}_{i}^{a}) + (1 - \delta_{\mathrm{mod}(t,\Delta T)}) \boldsymbol{f}^{a}(\boldsymbol{a}_{i}; \boldsymbol{p}_{i}^{a})] + \boldsymbol{g}^{a}(e_{i}^{h}, e_{i}^{w}, e_{i}^{m}) \tag{4a}$$

$$\dot{\boldsymbol{o}} = \boldsymbol{f}^{o}(\boldsymbol{o}; \boldsymbol{p}^{o}) + \boldsymbol{g}^{o}(\mathcal{P}^{o}\overline{e^{h}}, \mathcal{P}^{o}\overline{e^{w}}, \mathcal{P}^{o}\overline{e^{m}}, \mathcal{P}^{o}\boldsymbol{r}) \tag{4b}$$

$$\dot{\boldsymbol{l}} = \boldsymbol{f}^{l}(\boldsymbol{l}; \boldsymbol{p}^{l}) + \boldsymbol{g}^{l}(\mathcal{P}^{l}\overline{e^{h}}, \mathcal{P}^{l}\overline{e^{w}}, \boldsymbol{r}) \tag{4c}$$

$$\boldsymbol{a}_{s} = \sum_{i} \mathbf{W}_{i} \boldsymbol{a}_{i} \quad \text{if} \quad \delta_{\mathrm{mod}(t,\Delta T)} = 1, \tag{4d}$$

where $\boldsymbol{a}_{s}$ denotes the atmospheric state of the supermodel, $\mathbf{W}_{i}$ the diagonal matrices with weights on the diagonal for the $i$-th imperfect model, and the overbar indicates the weighted average over the models. At the instant times when a supermodel





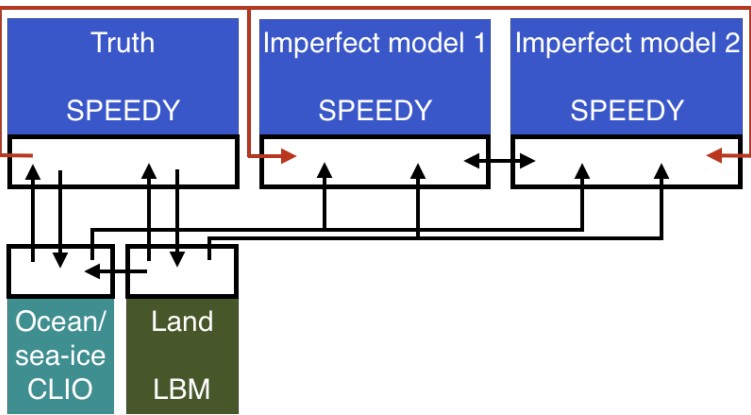

**Figure 2.** Schematic representation of the SPEEDO supermodel training.

is constructed (*i.e.*, $\delta = 1$), its state will be used to calculate the tendencies of the individual models. Otherwise, the individual
models just continue their runs without interacting.

## 3   Training methods

### 3.1   Training in SPEEDO

During the training for the supermodel based on SPEEDO, we regard the atmospheric model with standard parameter values
(specified later) as truth (Selten et al., 2017), whereas imperfect atmospheric models are created by perturbing those parameter
values. The ocean and the land models receive the heat, water and momentum fluxes from the perfect atmospheric model
only. All atmospheres receive the same information from the ocean and the land model, such that during training all imperfect
atmospheres only deviate from the observations due to their own difference, not because of the coupling with ocean and land
(see Fig. 2).

We follow a similar experimental setup as in the precursor study by Schevenhoven et al. (2019); in particular, to simulate
the imperfect models, we perturb the same parameters with the same values. These are the convection relaxation timescale,
the relative humidity threshold and the momentum diffusion timescale. The values used in the experiments are summarised in
Table 1.

The impact of perturbing parameters on the models' climate (*i.e.*, their long term behavior) is assessed on the basis of 40
years long simulations initiated on January 1st 2001. Table 2 shows the global mean average difference between the truth and
the imperfect models for different variables. We see that the imperfect models 1 and 2 have biases with opposite signs in all of
the variables. Note that their biases are comparable to those estimated for state-of-the-art global climate models (Collins et al.,
2013). The third model has biases in the same direction as model 1, but of generally larger amplitudes. We shall make use of
models 1 and 3 for the experiments with negative weights.



**Table 1.** Parameter values of perfect and imperfect models.

| model | convection relaxation timescale | relative humidity threshold | momentum diffusion timescale |
|---|---|---|---|
| perfect | 6 hours | 0.9 | 24 hours |
| model 1 | 4 hours | 0.85 | 18 hours |
| model 2 | 8 hours | 0.95 | 30 hours |
| model 3 | 3 hours | 0.75 | 14 hours |

**Table 2.** Global mean average difference between the imperfect models and the perfect model, calculated over the last 30 years of the simulation.

| model | temperature [$°C$] | precipitation [$mm/day$] | wind at 200 hPa [$m/s$] | wind at 850 hPa [$m/s$] | solar surface radiation [$W/m^2$] | cloudcover [%] |
|---|---|---|---|---|---|---|
| mod 1 | 1.37 | 0.11 | 1.04 | 0.07 | 2.06 | -1.59 |
| mod 2 | -0.38 | -0.04 | -0.31 | -0.03 | -1.13 | 0.87 |
| mod 3 | 3.20 | 0.26 | 2.25 | 0.03 | 3.95 | -3.37 |

Following Schevenhoven et al. (2019), we use global weights for both CPT and the synch rule. This means that we use the same weight for all grid points. By doing so we mitigate, and in the best case prevent, numerical instabilities. Note however that, different weights are allowed for each variable.

As long as there are enough observations to capture the global behaviour of the different models, spatially sparse observations are not expected to be an issue when constructing a weighted supermodel. Given that we focus here on the data sparsity in time, in the experiments we assume that all grid points are observed.

The prognostic variables exchanged between models are temperature, vorticity and flow divergence. The weights for the fluxes from atmosphere to ocean and to land are given by the average of the weights for the three prognostic variables. The SPEEDO time step during training is set to $\delta t = 15$ minutes.

The code for both training methods CPT and the synch rule is in the experiments in this paper integrated in the SPEEDO code. After the individual models have made their individual time steps, their states are exchanged between the models with coupling routines. Once all models have shared their knowledge, they can calculate the new supermodel state and the update of the weight according to the training method. The SPEEDO CPT and synch rule supermodel training code is available in Schevenhoven (2021).



## 3.2 Training with the synch rule

The synch rule was originally conceived for parameter optimization in Duane et al. (2007). We follow here a similar setting.
Let us assume that the parameters $\boldsymbol{q} \in \mathbb{R}^m$ appear linearly in the system for state variables $\boldsymbol{y} \in \mathbb{R}^n$, such that $\dot{\boldsymbol{y}} = \boldsymbol{f}(\boldsymbol{y}; \boldsymbol{q})$. The synch rule ensures convergence towards parameters $\boldsymbol{p} \in \mathbb{R}^m$ of the system for state variables $\boldsymbol{x} \in \mathbb{R}^n$, $\dot{\boldsymbol{x}} = \boldsymbol{f}(\boldsymbol{x}; \boldsymbol{p})$, provided that synchronization between the systems occurs if the parameters of both systems are equal: $\boldsymbol{y}(\boldsymbol{t}) \to \boldsymbol{x}(\boldsymbol{t})$ if $\boldsymbol{p} = \boldsymbol{q}$. The update of parameter $q_j$ for the $j$-th component of $\boldsymbol{q}$ reads:

$$\dot{\boldsymbol{x}} = \boldsymbol{f}(\boldsymbol{x}; \boldsymbol{p}) \tag{5a}$$

$$\dot{\boldsymbol{y}} = \boldsymbol{f}(\boldsymbol{y}; \boldsymbol{q}) - \mathbf{K}(\boldsymbol{y} - \boldsymbol{x}) \tag{5b}$$

$$\dot{q}_j = -\delta_j \sum_i e_i \frac{\partial f_i(\boldsymbol{y}, \boldsymbol{q})}{\partial q_j}, \tag{5c}$$

where $\boldsymbol{f} \in \mathbb{R}^n \to \mathbb{R}^n$ denotes the evolution function and $\mathbf{K}(\boldsymbol{y} - \boldsymbol{x})$ a connecting term between the two systems that nudges $\boldsymbol{y}$ towards $\boldsymbol{x}$. $\mathbf{K} \in \mathbb{R}^{n \times n}$ is a diagonal matrix of nudging coefficients, $\mathbf{K} = \mathrm{diag}(\boldsymbol{k})$. Furthermore, $e_i = y_i - x_i$ denotes the $i$-th component of the synchronization error, and $\delta_j$ an adjustable rate of learning scaling factor.

We have extended the use of the synch rule to the training of supermodels (Selten et al., 2017; Schevenhoven et al., 2019). In this context $\boldsymbol{q}$ refers to the supermodel weights, $\boldsymbol{y}$ to the supermodel state and $\boldsymbol{x}$ to the observations. The synch rule is initialized with certain values for $\boldsymbol{q}$ and during training the weights are updated according to the rule, such that the supermodel synchronizes with the observations. In order to keep the supermodel in the vicinity of the observations, the supermodel is nudged towards the observations by the term $\mathbf{K}(\boldsymbol{y} - \boldsymbol{x})$.

### 3.2.1 Nudging towards the observations

The sensitivity of the training results to the nudging strength $\mathbf{K}$ in SPEEDO was studied in Selten et al. (2017). It was found that an amount of $K = 1/24 \ \mathrm{hr}^{-1}$ nudging was sufficient to let identical SPEEDO models synchronize with a small error of less than $0.2°C$ between the models. Nevertheless, in the experiments with different versions of SPEEDO, the synchronization error increases by one order of magnitude. This amount of nudging is suitable for training, since it keeps the models close enough to the observations. Furthermore, a clear distinction can be made between an untrained and a well-trained supermodel in terms of the synchronization error between the supermodel and the observations. Because in our experiments nudging is applied only when observations are available, we found that a stronger nudging term than in Selten et al. (2017) is needed, and that its amplitude is approximately inverse related to the number of observations. We have some flexibility in the choice of the nudging strength in view of a certain insensitivity of the results. For instance, there is a range of values of $\mathbf{K}$ for which identical SPEEDO models synchronize to each other while between the different versions of SPEEDO a large error maintains. For the experiments in this paper $\mathbf{K}$ is therefore defined somehow arbitrarily, considering the fact that without nudging the error between models initially grows exponentially in time, but at some point saturates when the distance between the models is on





average as the distance between two random states on their attractors. Additionally, **K** is chosen equal for all the connected

variables temperature, vorticity and divergence.

## 3.3 Training with CPT

The CPT learning approach is based on an idea proposed by Smith (2001). CPT combines trajectories of different models dynamically, such that the solution space is virtually extended. The aim is to generate trajectories that follow the truth more closely. In Schevenhoven and Selten (2017), this idea has been developed into a supermodel training scheme.

The training phase of CPT starts from an observation. From the same initial state, the imperfect models run for a prede-fined "cross-pollination time", $\tau$, until an observation is available. The individual model predictions are then compared to the observation, and the model state that is closest to the observation will serve as initial condition for the next integration. In our experiments, the "closeness" to data is measured using the global Root Mean Squared Error (RMSE). In the case of a multi-dimensional (multivariate) model, such as SPEEDO, it is possible that at certain time steps different models are the

closest to the truth for different state variables. In this case, the initial condition for the next run is constructed by combining the portion of the state vector of each closest model state. This choice, while providing the closest-to-data initial condition, is prone to create imbalances in the model integration. Nevertheless, we experienced that as long as the update is global, such that each grid point receives the state of the same model for a certain variable, these imbalances are not a big issue. Otherwise, a possible solution is to use techniques from data assimilation (Carrassi et al., 2018) to make the initial condition suitable for

the individual models. An example is given in Du and Smith (2017) by using Pseudo orbit Data Assimilation (PDA).

    After the training, a CPT trajectory is obtained as a combination of different imperfect models, and we count how often each model has produced the best prediction of a particular component of the state vector during the training. These frequencies are then used to compute weights **W** for the corresponding states of the models. The superposition of the weighted imperfect model states forms the supermodel state. Since the frequency is used to compute the supermodel weights, the weights automatically

sum to 1, which is also functional to maintain physical balances.

### 3.3.1 The rationale behind CPT: an illustration

The CPT training method has been derived from a linear model assumption. Suppose we have two imperfect models with differential equations:

$$\dot{x}_1 = \alpha_1 \tag{6a}$$

$$\dot{x}_2 = \alpha_2, \tag{6b}$$

where $x_{1,2} \in \mathbb{R}$ state vectors and $\alpha_{1,2} \in \mathbb{Q}$ scalar direction coefficients. Assume the perfect model equations are given by:

$$\dot{x}_T = \alpha_T, \tag{7a}$$

where $x_T \in \mathbb{R}$ and $\alpha_T \in \mathbb{Q}$. Furthermore, assume the imperfect models complement each other such that $\alpha_1 < \alpha_T < \alpha_2$. Then there exists a convex combination $\alpha_1 \frac{n}{N} + \alpha_2(1 - \frac{n}{N}) = \alpha_T$, with $n, N \in \mathbb{N}$ and $n < N$. Choosing model 1 $n$ out of $N$ time





steps and $N - n$ times model 2 will result in a CPT trajectory that equals after $N$ time steps the perfect observation at that point in time. Constructing the CPT trajectory in such a way that always the model closest to the observations is chosen will result in an optimal trajectory.

Weather and climate models are chaotic instead of linear. Nevertheless, if enough observations are available, the model trajectory can at least partly be linearly approximated with a sufficient amount of observations $N$ such that weights can be
fine tuned. The obtained weights will not be perfect and possibly not as optimal as weights obtained with a cost function minimization approach. On the other hand, the results in Schevenhoven et al. (2019) show that the models are on short term linear enough to let the CPT approach work well. Moreover, CPT is a very fast method, and only few iterations are necessary as compared to the common approach of minimization of a cost function.

### 3.3.2    Duration of the training time

In Schevenhoven et al. (2019), the CPT training period in SPEEDO was set to one week. The time step in these experiments was 15 minutes and observations were available at every time step. Therefore the weights were based on a trajectory consisting of 672 time steps, a number that leads to a quite accurate estimation of the weights. In this work on the other hand, we set the maximal time between two subsequent observations to 24 hours, reducing the CPT trajectory to only seven steps in one week. Increasing the length of the training period is difficult because the supermodel trajectory may lose track of the observations
during training. To avoid this, the maximum duration for the training period is set to two weeks for $\Delta T = 24$hr. Moreover, we nudge towards the observations, as is done in the synch rule. To obtain more precise weights we use an iterative method as in Schevenhoven and Selten (2017).

### 3.3.3    Iterative method

In Schevenhoven and Selten (2017) an iterative method was proposed to obtain converged weights. The first iteration step gives
a first estimate of the weights of the supermodel. At the next iteration, the supermodel resulted from the previous iteration, is added as an extra imperfect model, and can thus potentially be the closest model to the observations. To calculate the new weights of the supermodel after the iteration, we adopt a simple linear approach. To see this, consider the case of two imperfect models, and assume that after an iteration the weights are $w_1^o$ for imperfect model 1 and hence $w_2^o = 1 - w_1^o$ for imperfect model 2. If the weights after the next iteration are $w_1^n$ for imperfect model 1, $w_2^n$ for imperfect model 2 and $1 - w_1^n - w_2^n$ for the
supermodel, then the new supermodel weights will be $w_1^n + w_1^o(1 - w_1^n - w_2^n)$ for imperfect model 1 and $w_2^n + w_2^o(1 - w_1^n - w_2^n)$ for imperfect model 2. The supermodel with these weights will replace the previous supermodel in the next iteration step. Ideally, the added supermodel is closer to the truth than the initial imperfect models. This can help to follow the observations for a longer period of time.





## 4 Results

### 4.1 Synch rule adaptations

In Schevenhoven et al. (2019) the synch rule, rewritten from Eq. (5), looked as follows:

$$\dot{W}_{i,j} \quad = \quad -\delta_j \, e_j f_{i,j}, \tag{8}$$

where $W_{i,j}$ denotes the weight of model $i$ for state variable $j$, $f_{i,j}$ the imperfect model tendency of model $i$ and state variable $j$, $e_j$ the synchronization error between the supermodel state and the observations, and $\delta_j$ an adjustable rate of learning scaling factor. This formula is derived without any prior assumption on the weights. In the context of noise free and continuously available observations, the weights turned out to sum approximately to one, which seems necessary in order to maintain physical balances. Nevertheless, when Eq. (8) is used in case of noisy and sparse in time observations, the weights do not sum to one anymore. If the deviation from one is too large, the supermodel state will be either too small or too large compared to the imperfect model states, possibly resulting in loss of synchronization with the observations and an even worse estimation of the next weight update.

We adapt the synch rule such that the weights are imposed to sum to one. This is achieved by using the tendency of the individual imperfect model $f_i$, but also by subtracting the equally weighted supermodel tendency $f_E = \frac{1}{N} \sum_{i=1}^{N} f_i$. The new synch rule is defined as (see Appendix A for a derivation):

$$\dot{W}_i \quad = \quad -\delta \, e(f_i - f_E), \tag{9}$$

where index $j$ is omitted to simplify the notation. From Eq. (9) it can be seen that the total update of the weights for the $N$ imperfect models equals zero: $\sum_{i=1}^{N} \dot{W}_i = -\delta \, e \sum_{i=1}^{N}(f_i - f_E) = -\delta \, eN(f_E - f_E) = 0$. Thus, if the initial weights sum to one, they will sum to one continuously throughout the training.

### 4.1.1 Adaptation to the nudging

Some adaptation is also needed in the nudging component. Too little nudging during training may lead to large errors between the imperfect models and the observations. In this case, the updates of the weights might go in a different direction than anticipated. The imperfect models and the observations might be in different phases, resulting in a converse sign of the synchronization error $e$. Interestingly, it is still possible to obtain converged weights in this case, only that the weights differ substantially from those obtained with more nudging towards the observations.

In the first experiment of Schevenhoven et al. (2019) the weights for temperature, vorticity and divergence all turned out to be around 0.3 for imperfect model 1 and 0.7 for imperfect model 2. We apply the same amount of nudging to the same imperfect models, except that the observations are available every second time step, instead of every time step. Then the weights converge to the weights given in Table 3.





**Table 3.** Weights for the supermodel trained by the synch rule with every second time step an observation available and the same amount of nudging towards the observations as in Selten et al., 2017; Schevenhoven et al., 2019. The weights are averaged over the last 10 weeks of training. Between brackets the standard deviation is given.

| model | T | VOR | DIV |
|---|---|---|---|
| model 1 | 1.15 (0.055) | 2.88 (0.070) | - 0.92 (0.046) |
| model 2 | - 0.15 (0.055) | - 1.88 (0.070) | 1.92 (0.046) |

When the weights converge towards stable values as in Table 3, the average update of the weights must equal to zero. Hence, at least one of the terms in Eq. (9) should be equal to zero on average. Since the imperfect models are not yet in equilibrium after one year (Schevenhoven et al., 2019), the average model tendency cannot be zero. This implies that the error between the supermodel and the observations must be equal to zero. However, a free run of 40 years with a supermodel with the weights from Table 3 results in a climatological error of up to $+2°$C in the northern hemisphere and up to $-2°$C in the southern hemisphere. Thus, too little nudging during training can result in a supermodel with a correct global average temperature (the opposite biases on the two hemispheres cancel), but very different dynamics compared to the observations.

There can also be too much nudging towards observations. In this case, a link with data assimilation can be made, where one has to find a middle ground between noisy observations and the model. Too much nudging towards the observations during training can result again in a converse sign of the synchronization error $e$. This leads to an incorrect update of the weights, making it more difficult to follow the observations during training.

### 4.2 CPT adaptations

For long training periods and/or noisy data, an iterative method as described in Sect. 3 might not be enough to let the CPT trajectory follow the observations adequately during training. A simple solution is to use a form of nudging towards the observations, similar to what is done in the synch rule. The equations for the CPT trajectory $\boldsymbol{x}_{CPT}$ with nudging, in an example with two imperfect models, are as follows:

$$\dot{\boldsymbol{x}}_1 = \delta_{\mathrm{mod}(t,\Delta T)}[\boldsymbol{f}(\boldsymbol{x}_{CPT},\boldsymbol{p}_1) + K(\boldsymbol{x}_{obs} - \boldsymbol{x}_{CPT})] + (1 - \delta_{\mathrm{mod}(t,\Delta T)})\boldsymbol{f}(\boldsymbol{x}_1,\boldsymbol{p}_1) \tag{10a}$$

$$\dot{\boldsymbol{x}}_2 = \delta_{\mathrm{mod}(t,\Delta T)}[\boldsymbol{f}(\boldsymbol{x}_{CPT},\boldsymbol{p}_2) + K(\boldsymbol{x}_{obs} - \boldsymbol{x}_{CPT})] + (1 - \delta_{\mathrm{mod}(t,\Delta T)})\boldsymbol{f}(\boldsymbol{x}_2,\boldsymbol{p}_2) \tag{10b}$$

$$\boldsymbol{x}_{CPT} = \begin{cases} \boldsymbol{x}_1, \text{ if } \|\boldsymbol{x}_1 - \boldsymbol{x}_{obs}\| \le \|\boldsymbol{x}_2 - \boldsymbol{x}_{obs}\| \text{ and } \delta_{\mathrm{mod}(t,\Delta T)} = 1 \\ \boldsymbol{x}_2, \text{ if } \|\boldsymbol{x}_2 - \boldsymbol{x}_{obs}\| \le \|\boldsymbol{x}_1 - \boldsymbol{x}_{obs}\| \text{ and } \delta_{\mathrm{mod}(t,\Delta T)} = 1, \end{cases} \tag{10c}$$

where $\boldsymbol{x}_{obs}$ denote the observations, $\delta$ the Kronecker delta, $\Delta T$ the observation frequency and $K$ the nudging coefficient. In the experiments in this section $K$ is equal for all state variables. As with the synch rule, the nudging strength needs to be enough to follow the observations, but it should not be too strong. The goal of CPT is to see how models can compensate for each other. Therefore, deviations from the original observations can be advantageous, as long as there are imperfect models





able to counteract this deviation. Nudging the imperfect model states to a value very close to the observations will lead to a too frequent choice of the model that is on average closest to the observations, thus limiting the diversity of representation within the supermodel.

### 4.3 Limitations of sparse and noisy observations

In this section we assess to which extent observations can be noisy and sparse in time before the CPT or the synch rule training methods are no longer able to produce weights close to optimal. To systematically evaluate this, we choose 4 different observation frequencies $\Delta T$: 15 minutes, 1 hour, 6 hours and 24 hours. Since for the standard CPT training time of one week the weights for $\Delta T = 24$hr would only be based on 7 steps, the training time for this observation frequency is doubled to two weeks. The error in the observations is unbiased and Gaussian distributed $\sim N(0, \sigma)$, where standard deviation $\sigma$ is chosen

to be equal to either $0.5\%, 2.5\%$ or $5\%$ of the spatial standard deviation $\sigma_X$ of the observations per prognostic variable $X$. Hence $\sigma_X = \frac{1}{N}\sqrt{\sum_i (X_i - \overline{X})^2}$, with $i$ ranging over all $N = 96 \times 48 \times 8$ gridpoints and $\overline{X}$ denoting the spatial mean value. For temperature this corresponds to a standard deviation of $\sim 0.15°C$, $0.75°C$ and $1.5°C$. Table 4 denotes the chosen nudging coefficient $K$ and the resulting weights together with their variance. For the experiments with $\Delta t = 15$ minutes the same nudging strength $K$ is chosen as in Schevenhoven et al. (2019): $K = 1/24\ \text{hr}^{-1}$, which corresponds to $K = 1\ \text{day}^{-1}$. All CPT

experiments are performed with the iterative method.

Figure 3 shows the weights from Table 4 in one plot such that the differences between the methods become clear. The horizontal lines (continuous for model 1 and dashed for model 2) indicate the weights obtained by CPT and the synch rule in Schevenhoven et al. (2019), in which case the observations were perfect and available at every time step. Despite the optimal

weights for $\Delta T = 15$ minutes are not necessarily expected to be optimal for, *e.g.* $\Delta T = 24$ hours, in this particular experiment, the two cases show similar weights. From Fig. 3a and Fig. 3b it can be seen that if observations are available at each time-step ($\Delta T = 15$ minutes), the synch rule gives slightly better results for noisier observations than the CPT scheme. For the synch rule, the weights turn out to be almost exactly the same for all three levels of noise. For $\Delta T = 1$ hour, the results for CPT and the synch rule seem similar for the two lowest noise levels. For the highest noise level, both methods seem to struggle a bit

more to obtain good weights, the models are more equally weighted. Decreasing the observation frequency further to $\Delta T = 6$ hours results in the same pattern, CPT performing slightly better. Once good CPT weights have been found, they remain very consistent throughout the year, indicated by the standard deviation of 0. For the largest observation window $\Delta T = 24$ hours, again the synch rule seems to encounter somewhat more difficulties in following the temperature observations for the highest noise level. Overall however, both methods perform well in the context of sparse and noisy observations.

### 345 4.4 Negative weights

Imperfect model 1 and 2 complement each other in important physical variables such as temperature and wind. Model 1 tends to overestimate their global average values, while model 2 underestimates them. Together they form a convex hull (Schevenhoven and Selten, 2017), which results in positive supermodel weights. On the other hand, using model 1 and model 3 to construct a





**Table 4.** Weights for the supermodel trained by CPT and the synch rule. Between brackets, the standard deviation over the year (CPT) or the standard deviation over the last 10 weeks of training (synch rule) is given. The weights are only given for model 1, for model 2 the weight equals 1 - weight of model 1.

$\Delta T = \textbf{15 minutes}$

| method | noise [%] | $K$ [day$^{-1}$] | T | VOR | DIV |
|---|---|---|---|---|---|
| **CPT** | 0.5 | 0.0 | 0.32 (0.016) | 0.40 (0.010) | 0.29 (0.032) |
| | 2.5 | 1.0 | 0.34 (0.015) | 0.39 (0.005) | 0.31 (0.031) |
| | 5.0 | 1.0 | 0.39 (0.020) | 0.39 (0.010) | 0.28 (0.031) |
| **synch** | 0.5 | 1.0 | 0.33 (0.012) | 0.39 (0.005) | 0.35 (0.004) |
| | 2.5 | 1.0 | 0.32 (0.006) | 0.39 (0.003) | 0.34 (0.003) |
| | 5.0 | 1.0 | 0.33 (0.016) | 0.39 (0.005) | 0.34 (0.006) |

$\Delta T = \textbf{1 hour}$

| method | noise [%] | $K$ [day$^{-1}$] | T | VOR | DIV |
|---|---|---|---|---|---|
| **CPT** | 0.5 | 0.0 | 0.35 (0.011) | 0.40 (0.004) | 0.35 (0.009) |
| | 2.5 | 20.0 | 0.34 (0.011) | 0.38 (0.000) | 0.36 (0.008) |
| | 5.0 | 20.0 | 0.43 (0.015) | 0.40 (0.005) | 0.40 (0.013) |
| **synch** | 0.5 | 20.0 | 0.30 (0.006) | 0.39 (0.001) | 0.36 (0.002) |
| | 2.5 | 20.0 | 0.35 (0.005) | 0.39 (0.001) | 0.39 (0.004) |
| | 5.0 | 20.0 | 0.46 (0.005) | 0.39 (0.002) | 0.46 (0.006) |

$\Delta T = \textbf{6 hours}$

| method | noise [%] | $K$ [day$^{-1}$] | T | VOR | DIV |
|---|---|---|---|---|---|
| **CPT** | 0.5 | 0.0 | 0.32 (0.000) | 0.43 (0.000) | 0.32 (0.030) |
| | 2.5 | 20.0 | 0.31 (0.008) | 0.39 (0.000) | 0.32 (0.000) |
| | 5.0 | 40.0 | 0.46 (0.054) | 0.36 (0.010) | 0.39 (0.000) |
| **synch** | 0.5 | 40.0 | 0.33 (0.003) | 0.39 (0.002) | 0.33 (0.007) |
| | 2.5 | 40.0 | 0.36 (0.013) | 0.39 (0.009) | 0.34 (0.012) |
| | 5.0 | 40.0 | 0.48 (0.016) | 0.42 (0.011) | 0.42 (0.006) |

$\Delta T = \textbf{24 hours}$

| method | noise [%] | $K$ [day$^{-1}$] | T | VOR | DIV |
|---|---|---|---|---|---|
| **CPT** | 0.5 | 20.0 | 0.38 (0.020) | 0.43 (0.000) | 0.29 (0.000) |
| | 2.5 | 70.0 | 0.29 (0.000) | 0.29 (0.000) | 0.27 (0.011) |
| | 5.0 | 70.0 | 0.40 (0.057) | 0.43 (0.051) | 0.39 (0.085) |
| **synch** | 0.5 | 70.0 | 0.37 (0.012) | 0.42 (0.017) | 0.38 (0.010) |
| | 2.5 | 70.0 | 0.37 (0.010) | 0.42 (0.011) | 0.38 (0.005) |
| | 5.0 | 70.0 | 0.48 (0.013) | 0.40 (0.036) | 0.36 (0.008) |



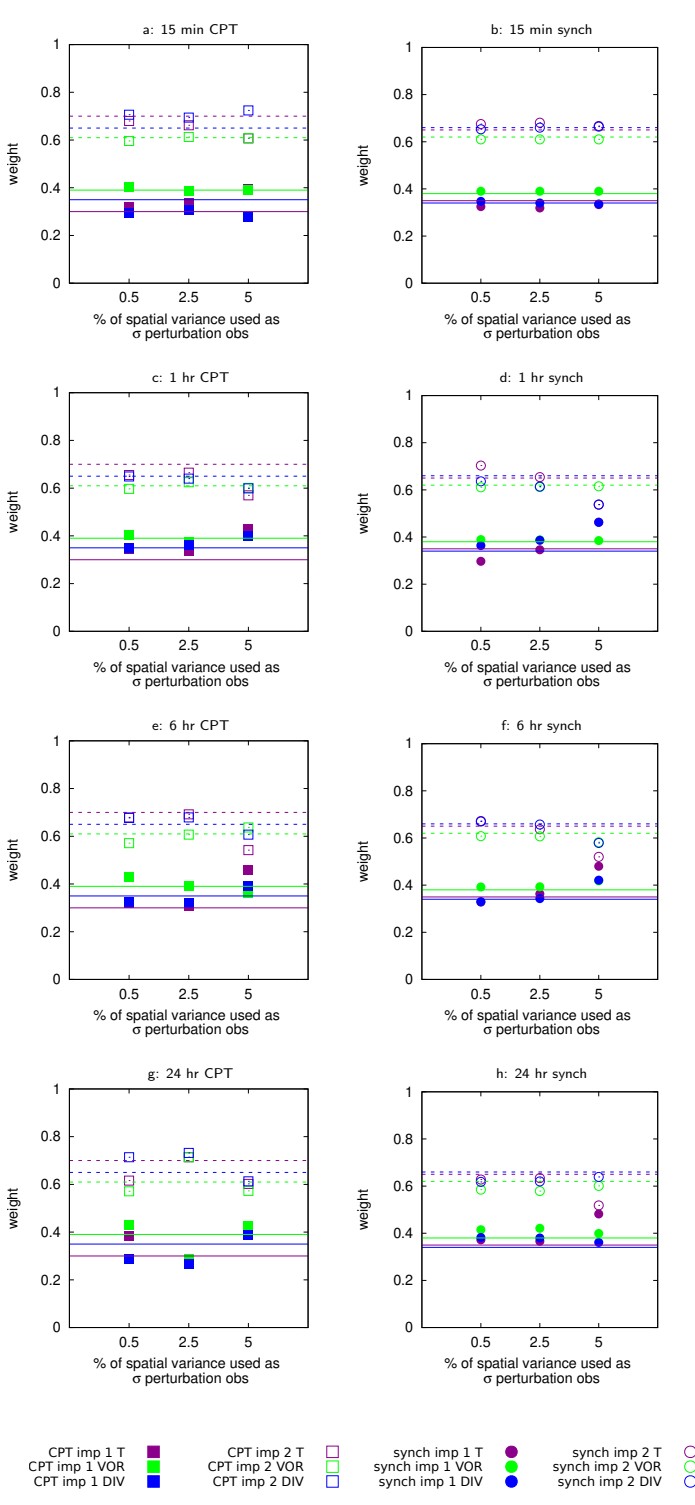

**Figure 3.** Weights for the supermodel trained by CPT and the synch rule. The horizontal lines (continuous mod 1, dashed mod 2) indicate the weights obtained by CPT and synch rule training in Schevenhoven et al. (2019), in which case the observations were perfect and available at every time step.





supermodel implies the need for negative weights. The synch rule naturally allows negative weights, since we did not impose

any restrictions on the weights. In Schevenhoven et al. (2019) synch rule training has been performed with model 1 and model 3, resulting in a supermodel with partly negative weights that outperformed both imperfect models in short and long term forecast quality.

CPT training does not automatically produce negative weights, since the weights are based on the frequency by which the imperfect models are chosen. Nevertheless, CPT training can give negative weights too, although with boundary restrictions.

In the standard CPT training, one chooses whether one of the imperfect models is the closest to the observations, or in addition, whether the supermodel is closest in the iterative method. To obtain negative weights one can also choose a predefined combination of the imperfect models, for example: $\boldsymbol{x}_{neg1} = \alpha \boldsymbol{x}_1 + (1-\alpha)\boldsymbol{x}_3$, with $\alpha \in \mathbb{R}$. If one defines an additional predefined combination $\boldsymbol{x}_{neg2} = (1-\alpha)\boldsymbol{x}_1 + \alpha \boldsymbol{x}_3$, the range for weights $\boldsymbol{w}_1$ and $\boldsymbol{w}_3$ for imperfect model 1 and 3 is between $\alpha$ and $1-\alpha$ if $\alpha \notin [0,1]$.

In this experiment we choose $\alpha = -1$, such that $\boldsymbol{w}_1, \boldsymbol{w}_3 \in [-1,2]$. The experiment is the same as in Schevenhoven et al. (2019): every time step an observation is available, every time step either $x_{neg1}$ or $x_{neg2}$ per variable is chosen as closest model state and the training period is one week. Table 5 shows the weights and associated variance of the weights. The weights are remarkably similar to the weights of the synch rule experiment with negative weights in Schevenhoven et al. (2019). The weights for vorticity and divergence differ by 0.16, the weights for temperature by only 0.03. The statistics of a 40-year

**Table 5.** Weights for the supermodel trained by CPT allowing for negative weights. Between brackets the standard deviation over the year is given.

| model | T | VOR | DIV |
|---|---|---|---|
| model 1 | 1.33 (0.028) | 1.84 (0.058) | 0.56 (0.079) |
| model 3 | - 0.33 (0.055) | - 0.84 (0.058) | 0.44 (0.079) |

supermodel run with the weights from Table 5 are therefore quite similar to the climatology of the supermodel in Schevenhoven et al. (2019). Table 6 shows that the supermodel outperforms both imperfect models in temperature, precipitation, wind, cloud cover and surface solar radiation compared to the values in Table 2.

**Table 6.** Global mean average difference between the supermodel with negative weights and the perfect model, calculated over the last 30 years of the simulation.

| model | temperature [$^\circ C$] | precipitation [$mm/day$] | wind at 200 hPa [$m/s$] | wind at 850 hPa [$m/s$] | solar surface radiation [$W/m^2$] | cloudcover [%] |
|---|---|---|---|---|---|---|
| super | 0.62 | 0.10 | 0.57 | -0.02 | 1.43 | -0.87 |





## 5 Discussion and conclusion

We have shown the potential of the CPT and synch rule training methods to train a weighted supermodel on the basis of
noisy and sparse in time observations. The CPT training method is based on "crossing" different model trajectories and thus
generating a larger ensemble of possible trajectories. The synch rule adapts the weights to the individual models on-the-fly
during the training, such that the supermodel synchronizes with the observations. In our previous work (Schevenhoven et al.,
2019), it was shown that both methods were able to improve weather and climate predictions in a noisy-free and highly frequent
observational setting, using different parametric versions of the global coupled atmosphere–ocean–land model SPEEDO. In this
study, we moved towards realism by handling the case of noisy data that are not available at each of the models' computational
time step. We have generated synthetic noisy observations by adding zero-mean Gaussian noise, with variance as large as
1.5°C in temperature. These synthetic noisy observations are made available at different intervals, of 1 hr, 6 hrs or 24 hrs.
Both methods needed adaptations over the original formulations given in Schevenhoven et al. (2019) in order to train the
weighted supermodel on the basis of noisy and sparse in time observations. The new variants of the training methods have
proved robustness against these changes in the observational scenario and shown capabilities to give adequate weights.

To handle noisy and sparse in time data, we use nudging in both methods: this choice showed to be pivotal to ensure correct
updates of the weights. For the synch rule the nudging strength was increased while for CPT the nudging term was not present
in the original formulation and has been introduced here.

For the synch rule it is necessary that the sum of the weights remains equal to one in order to maintain physical balances. In
the noise-free framework of Schevenhoven et al. (2019), this is ensured automatically. Nevertheless, in the current framework
we had to impose the condition that the weights sum to one, which is achieved by subtracting the equally weighted tendency
term in the synch rule equation. Besides the inclusion of nudging in the CPT method, the use of an iterative approach within
CPT further helped to keep track of the data. Additionally, in Schevenhoven et al. (2019) the synch rule was able to produce
negative weights in case the imperfect models cannot compensate for each other's biases with positive weights. In this paper,
we have gone beyond this and developed a method to obtain negative weights also within the CPT method.

CPT and the synch rule both update the weights based on the difference between the model trajectories and the observations,
and on the difference between the imperfect model tendencies. Despite using similar ingredients, CPT and the synch rule give
different results for sparse and noisy observations. In particular, the synch rule trajectory seems to diverge slightly earlier from
the observations than the CPT one. A possible reason stands on the different use of the models' tendencies. With CPT, the
imperfect models run unconstrained from the data in the period, $\Delta T$, between subsequent observations. If $\Delta T$ is large enough
the model trajectories will have a large spread before being compared at the next observation time. Choosing the right model
from this large spread can quickly reduce the distance to the observations. For the synch rule, on the other hand, one integrates
the supermodel instead of the individual models as in CPT. Once the synch rule supermodel trajectory has diverged from
the observations, it can be more difficult to get back to the observations compared to the CPT training, since the supermodel
weights need to be adapted such that the next integration period will bring the supermodel closer to the observations. If $\Delta T$



is small, CPT and the synch rule are very comparable methods, as we have also seen in the results of the negative weight experiment in Sect. 4.

**Future directions**

Despite the application of the iterative method and of the nudging, CPT and synch rule may still struggle to stay very close
to the observations. To increase the chance to obtain a proper trajectory, one could work with an augmented ensemble of trajectories. This ensemble could consist of trajectories starting from slightly different initial conditions, or trajectories that emerge from a model nearby the closest model to an observation. One could make a comparison with the particle filter method (see *e.g.* van Leeuwen et al. (2019)), where trajectories that do not fall within the likelihood of the observations are pruned and one continues with the trajectories within the likelihood of the observations from slightly perturbed initial conditions. In our
case, the best trajectory after training can be obtained by comparing the RMSE between the trajectories and the observations.

Both training methods seem in principle more suitable for short rather than longer timescales, since for both training rules it is important that the imperfect models stay close enough to the observations. For longer timescales this can be difficult. Despite the action of the nudging the models can be out of the data phase as long as time evolves. If the observations are lost, the 'closest' model in the CPT training is not necessarily the one that contributes most to improving the supermodel dynamics.
If during synch rule training the supermodel loses the observations, a new, non-optimal equilibrium for the weights can be found, as we have seen in Sect. 4. Having said this, still both methods could be useful if one prefers to combine models only on a seasonal or even longer timescale (under the assumption that with this limited amount of exchange the models are still synchronized to some extent). For both CPT and the synch rule, one can average over the observations to potentially obtain a correct sign whether the supermodel is either over- or underestimating the observations.

Until now the distance between models and model-to-data has been the RMSE. If one is training a supermodel with improved skill on longer timescales, it is possible that the appearance of specific climatological features of the models is of more importance than a small RMSE. In that case the distance between observations and models can be defined in a different way. For example, if the imperfect models suffer from an erroneous double ITCZ, one can increase the weight of the model which is on average closer to a single ITCZ. Additionally, one can define different weights for different periods of time, for example
seasonally dependent weights. Despite these possibilities in adapting the training methods, there are some conditions that need to be fulfilled when CPT or the synch rule are used on longer timescales. The methods only work if the models can compensate for each other. For example, when both models have been spun up for a sufficient amount of time and are stable in state space, both CPT and the synch rule cannot give useful weights. In the case of CPT, the model that is on average closest to the observations will be chosen repeatedly. For the synch rule, the average model tendency will be zero over a sufficient amount
of time, hence there will be no update of the weights on average over time. Therefore, for both training methods the imperfect models cannot already reside on their own attractor, the tendency towards their attractor needs to be visible.

To make the training methods suitable for state-of-the-art models it needs to be taken into account that state-of-the-art models can differ in grid point resolution and time steps. In this paper, for both CPT and the synch rule during training the imperfect model states are replaced, in the case of the synch rule the imperfect model states are replaced by the new supermodel state, and





in the case of CPT the imperfect model states are replaced by the state of the closest model. To apply the training methods in state-of-the-art models, techniques from data assimilation can be used to combine the states in a dynamical consistent manner (Carrassi et al., 2018). With the use of these techniques both CPT and the synch rule in principle seem to be suitable for training a state-of-the-art supermodel.

**Appendix A: The synch rule in case of noisy and sparse observations**

The general form of the synch rule as given in Duane et al. (2007) equals:

$$\dot{q}_j = -\delta_j \sum_i e_i \frac{\partial f_i(\boldsymbol{x}, \boldsymbol{q})}{\partial q_j}, \tag{A1}$$

where $q_j$ is the updated parameter, $\delta_j$ the adaptation rate, $e_i$ the difference between the truth and model $i$ and $f_i$ the time derivative of model $i$ for the particular parameter. For derivation, see Duane et al. (2007). In our case $q_j$ are the weights of the supermodel with respect to a certain variable $j$ and model $i$ is the supermodel. Hence we can rewrite it to:

$$\dot{q}_{kj} = -\delta_j e_j \frac{\partial \boldsymbol{f}_s(\boldsymbol{x}, \boldsymbol{q})}{\partial q_{kj}}, \tag{A2}$$

where $k$ denotes imperfect model $k$, and $q_{kj}$ the weight for model $k$ corresponding to variable $j$. The error $e_j$ is the difference between the truth and the supermodel, and $\boldsymbol{f}_s$ is the time derivative of the supermodel. At this point we can make a choice. We can write $\boldsymbol{f}_s$ just as a superposition of imperfect models $k$ as done in Schevenhoven et al. (2019):

$$\boldsymbol{f}_s = \sum_i \mathbf{W}_k \boldsymbol{f}_k, \tag{A3}$$

without any constraints on the weights. Then the term $\frac{\partial \boldsymbol{f}_s(\boldsymbol{x}, \boldsymbol{q})}{\partial q_{kj}}$ is just the time derivative of the imperfect model $k$ corresponding to variable $j$ implying we can simplify the rule further to (Schevenhoven et al., 2019):

$$\dot{W}_{kj} = -\delta_j e_j f_{kj}. \tag{A4}$$

Although no constraint was imposed on the weights, they automatically turned out to sum to one in Schevenhoven et al. (2019). In Schevenhoven et al. (2019) a partial explanation is given based on maintaining physical balances. The observations to train the weights on in this paper were noise-free and available every time step. If we remove these assumptions, the update of the weights might disturb the physical balances and therefore the synchronization of the supermodel with the observations could be lost, resulting in meaningless weights. A possible solution is to impose that the weights sum to one. In an example of only two imperfect models, the supermodel time derivative can be written as follows for weight $\mathbf{W}_1$:

$$\boldsymbol{f}_s = \mathbf{W}_1 \boldsymbol{f}_1 + (\mathbf{1} - \mathbf{W}_1) \boldsymbol{f}_2 \tag{A5}$$

Then rewriting Eq. (A4) omitting variable $j$ for simplicity gives:

$$\dot{W}_1 = -\delta\, e (f_1 - f_2). \tag{A6}$$



To get the equation for the update of $\mathbf{W}_2$ as well from the formula for $\boldsymbol{f}_s$, we can rewrite $\boldsymbol{f}_s$ to:

$$\boldsymbol{f}_s = \frac{1}{2}\Big(\mathbf{W}_1\boldsymbol{f}_1 + (\mathbf{1}-\mathbf{W}_1)\boldsymbol{f}_2\Big) + \frac{1}{2}\Big((\mathbf{1}-\mathbf{W}_2)\boldsymbol{f}_1 + \mathbf{W}_2\boldsymbol{f}_2\Big) \tag{A7}$$

Then the equations are:

$$\dot{W}_1 = -\widetilde{\delta}\, e(f_1 - f_2) \tag{A8}$$

$$\dot{W}_2 = -\widetilde{\delta}\, e(f_2 - f_1). \tag{A9}$$

So the total update of the weights $W_1$ and $W_2$ is equal to zero, hence initialization with weights that sum to one will result in weights that remain summed to one. This rule can be further generalized. Rewrite for $N$ models $\boldsymbol{f}_s$ to:

$$\boldsymbol{f}_s = \frac{1}{N}\sum_{i=1}^{N}\Big(\Big[(\mathbf{1}-\sum_{k,k\neq i}\mathbf{W}_k)\boldsymbol{f}_i\Big] + \sum_{j\neq i}^{N}\mathbf{W}_j\boldsymbol{f}_j\Big) \tag{A10}$$

Taking the derivative of $\boldsymbol{f}_s$ with respect to weight $\mathbf{W}_i$, $\frac{\partial \boldsymbol{f}_s(\mathbf{x},\mathbf{W})}{\partial \mathbf{W}_i}$, results in the following learning rule:

$$\dot{W}_i = -\delta\, e((N-1)f_i - \sum_{k,k\neq i} f_k), \tag{A11}$$

which can be rewritten to:

$$\dot{W}_i = -\widetilde{\delta}\, e(f_i - \frac{1}{N}\sum_{k} f_k), \tag{A12}$$

where $\frac{1}{N}\sum_{k} f_k)$ can simply be written as $f_e$, with $e$ meaning equally weighted, since $f_e$ is the equally weighted supermodel tendency, so

$$\dot{W}_i = -\widetilde{\delta}\, e(f_i - f_e). \tag{A13}$$

From Eq. (A13) it can also easily be seen that the total update of the weights equals zero: $\sum_{i=1}^{N}\dot{W}_i = -\widetilde{\delta}\, e\sum_{i=1}^{N}(f_i - f_e) = -\widetilde{\delta}\, eN(f_e - f_e) = 0$.

*Code availability.* The exact version of the SPEEDO model code with the CPT and synch rule training integrated that is used to produce the results used in this paper is archived on Zenodo (Schevenhoven, 2021), as are input data and scripts to run the model for all the simulations presented in this paper.

*Author contributions.* FS conceived the study, carried out the research and led the writing of the manuscript. AC provided input for the interpretation of the results and the writing.

*Competing interests.* The authors declare that they have no conflict of interest.



*Acknowledgements.* AC has been funded by the UK Natural Environment Research Council award NCEO02004.



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
