# Peer review of "Training a supermodel with noisy and sparse observations: a case study with CPT and the synch rule on SPEEDO - v.1"

_Geoscientific Model Development, 2021_

## Author Comment (AC1)

**Response to Reviewers**

Francine Schevenhoven[1,2,3] and Alberto Carrassi[4,5]

[1]Geophysical Institute, University of Bergen, Bergen, Norway
[2]Bjerknes Centre for Climate Research, Bergen, Norway
[3]University of Colorado, Boulder, USA
[4]Dept. of Meteorology and NCEO, University of Reading, United Kingdom
[5]Department of Physics and Astronomy "Augusto Righi", University of Bologna, Bologna, Italy

We are thankful to both Reviewers for their careful reading of our manuscript and their valuable suggestions and points of criticism. We have tried to address them all and provide details point-by-point below.

The Reviewer comments are reported in normal font, our replies are written in italics and preliminary proposed changes to the manuscript are denoted between brackets.

**Reviewer 1**

The paper applied two approaches, CPT and synchronization based, to "combine" multiple imperfect models to form a so called supermodel to noisy and sparse observations. The general concept is quite interesting and potentially could be very valuable to multi-model forecasting. The authors inherited previous works on CPT and synch rule, and relaxed noise-free assumption, proposed adaptations and demonstrated in SPEEDO experiments. Overall I think this paper is publishable with some revisions. Please find my concerns listed below.

1. It seems that the "imperfect" scenario described in the paper (for example section 2) are limited in perfect model class with imperfect model parameters. Is the proposed methodology applicable to a more general scenario where the model class is imperfect, i.e. there is no parameter value would lead to a perfect model.

   *The imperfect models in this study are indeed different from the "perfect" model only in terms of different parameter values. While this is only a subset (and maybe the easiest) of all possible sources of model error, the current work is a first step in applying the supermodel methodology. However, supermodeling is not restricted to this case and its general aim is to combine models of different natures, including differences in resolutions, parametrizations and in the resolved scales. The ultimate goal of supermodeling is to combine state-of-the-art models, for example models that are currently used by the IPCC. Nevertheless, the model combination requires defining a common (observed) space in which the individual model states can be combined into a supermodel state. This implies critical choices on how to project the combined states to each individual model space and, vice versa, the individual states to the combined model space. In our work, we intentionally placed ourselves in the context of parametric error to avoid the aforementioned complications and to allow focusing on the main object of the study, which is the supermodel training method. In the last paragraph of Sect. 5 we briefly explain how we technically would make the step towards relaxing this assumption.*

2. I am strongly against the idea (Line 28,44) that the benefit of using MME is due to "errors tend to cancel each other", this misinterpretation is a result of only considering the ensemble mean. Note that the purpose of using the ensemble forecasts (Leutbecher and Palmer (2008)) is to account for uncertainty.

*The Reviewer is right and we thank her/him for raising this point. In the revised version of the manuscript, we will make clear that the main purpose of using ensemble forecasts is to account for uncertainty.*

*"[The MME is most importantly a very powerful and useful approach to account for uncertainty. Furthermore it is possible to achieve better statistics such as the mean; this is because errors tend to cancel each other (Hagedorn et al., 2005). Generally, the models are equally weighted in a MME mean, as is the case for, e.g. the CMIP runs in the IPCC reports.]"*

3. The authors proposed the possibility of negative weights when form the supermodel. Personally I am a little concern about this approach, especially how would one interpret the negative weights? Can we first bias correct the models (for example Line 158, models 1 and before combine them?

*We admit that naming them "weights" is potentially confusing when they get negative values. In fact, those have rather to be seen as coefficients of a (potentially the best) combination of multiple models. We opted to keep calling them "weights" for consistency with our previous works and with the literature on supermodeling. Negative weights/coefficients help when the individual models have errors with the same sign, for example when they all overestimate temperature. By using negative weights, the dynamics of the resulting supermodel could be made more consistent to data than the individual models. In alternative, or in addition, one could indeed apply a pre-bias correction as suggested by the Reviewer and then use the bias-corrected model to construct the supermodel. This is certainly a relevant venue to pursue and such a de-biasing pre-step might be mandatory when working with more complex models. Nevertheless, there will never be guarantees that the bias could be fully removed, and thus the need for negative weights still holds. It is finally worth stressing that bias correcting the model is a statistical rather than a dynamical approach, that can co-exist with the dynamically-based construction of a supermodel. It would be interesting to combine the two in a future work.*

4. It is a little unclear what exactly is a supermodel. is the sentence at (Line 35) a formal definition of the "supermodel"?

*The essence of a supermodel is that models exchange information with each other during the simulation. They can achieve this by sharing their tendencies, as is mentioned in Line 35, but there are other methods as well. In this paper we define a weighted supermodel, in which models share their states, in order to be combined in a weighted supermodel state. We will adjust the text to clarify this point.*

*"[Along this line, in the supermodel approach, models are combined during the simulation by sharing their own tendencies or states with each other, and not just their outputs as with the MME.]"*

5. Line 98-99, it seems that achieving a synchronized state is a good thing, please define what is a synchronized supermodel state and clarify why it is desirable.

*By synchronization we mean that the different model trajectories are correlated in time, in contrast to independent model*

*runs. In a supermodel the models influence each other such that (at least part of) the physical processes are in phase.*

60 *This is desirable because synchronization creates a new trajectory that is a "solution" of the supermodel. In Line 40-45 of the introduction this concept is explained. We will clarify the sentence in Line 98-99.*

*"[In fact, combining the model tendencies every $\Delta T > \delta t$ can result in a much less synchronized supermodel state, thus possibly leading to a supermodel with poor forecast skill since the resulting supermodel trajectory from models that are not well enough synchronized will suffer from variance reduction and smoothing.]"*

6. Line 87, "observations were available", this statement needs to be clarified as the observations were not mention early
65     or in the equations 1a-c.

*The observations refer to the "historical observations" on which basis the supermodel is trained. This was mentioned in the second but last paragraph of the introduction (Line 58-63).*

7. Section 3.3, please give more details about how the CPT trajectories are generated.

*Full and detailed explanations on how the CPT trajectories are generated are given in Schevenhoven and Selten (2017).*
70 *In the manuscript we provide a succinct, but hopefully sufficient, summary and in the revised manuscript we will refer to Schevenhoven and Selten (2017) more explicitly.*

*"[The training phase of CPT starts from an observation. From the same initial state, the imperfect models run for a predefined "cross-pollination time", $\tau$, until an observation is available. The individual model predictions are then compared to the observation, and the model state that is closest to the observation will serve as initial condition for the*
75 *next integration. In our experiments, the "closeness" to data is measured using the global Root Mean Squared Error (RMSE). In the case of a multi-dimensional (multivariate) model, such as SPEEDO, it is possible that at certain time steps different models are the closest to the truth for different state variables. In this case, the initial condition for the next run is constructed by combining the portion of the state vector of each closest model state. This choice, while providing the closest-to-data initial condition, is prone to create imbalances in the model integration. Nevertheless, we*
80 *experienced that as long as the update is global, such that each grid point receives the state of the same model for a certain variable, these imbalances are not a big issue. Otherwise, a possible solution is to use techniques from data assimilation (Carrassi et al., 2018) to make the initial condition suitable for the individual models. An example is given in Du and Smith (2017) by using Pseudo orbit Data Assimilation (PDA). After the training, a CPT trajectory is obtained as a combination of different imperfect models, and we count how often each model has produced the best prediction of*
85 *a particular component of the state vector during the training. These frequencies are then used to compute weights $\mathbf{W}$ for the corresponding states of the models. The superposition of the weighted imperfect model states forms the supermodel state. Since the frequency is used to compute the supermodel weights, the weights automatically sum to 1, which is also functional to maintain physical balances. See Schevenhoven and Selten (2017) for a more extensive and figurative explanation of the CPT training scheme. ]"*

8. Paragraph at Line 240, It seems that the authors suggest that when the models are linear, the CPT approach works well. Why it doesn't if the model is highly nonlinear?

*We apologise for not having clarified this better in the original version of the manuscript. The key is not the dynamical nature of the models, i.e. whether they are linear or nonlinear, but the tradeoff between the data sampling time and the regime of evolution of the differences among the individual model trajectories in between subsequent data times. In our study, the observations are frequent and accurate enough such that between two subsequent observation times the difference between the models grows quasi-linearly or weakly nonlinearly. Hence the indicator of the chance of a successful training, is the combination of the dynamics of the difference between the imperfect models, and the frequency and accuracy of the data. We will clarify this in the text. To which extent the CPT approach works well in more complex highly nonlinear models is still an open question, but is expected to be generally determined by the interplay of the aforementioned factors. The results so far suggest that CPT could be suitable to train complex climate supermodels.*

*"[Weather and climate models are nonlinear and often chaotic. The key to success is however not the dynamical nature of the models, i.e. whether they are linear or nonlinear, but the trade-off between the data sampling time and the regime of evolution of the differences among the individual model trajectories in between subsequent data times. If enough observations are available during training, the difference between the imperfect models between subsequent observation times can be described as quasi-linear, therefore still making it possible for the CPT training to work well. ]"*

9. Table 4 and Figure 3 shows the weights for supermodel trained by CPT and synch rule, it would be interesting to see the forecast performance of CPT compared with synch rule.

*The resulting weights of this study obtained by sparse and noisy observations seem quite comparable to supermodel weights obtained with perfect, noise-free observations. We therefore expect that the forecast quality of the supermodels will be similar as well. Following the Reviewer's remark, we show in Fig. 1 below, the Root-Mean-Squared-Error (RMSE) of supermodels based on noisy-sparse observations (s-CPT and s-synch noisy obs; with $\Delta T = 24hr$ and $\sigma = 5\%$) and of those based on perfect data (named - perf obs), for a 2 weeks forecast. See Table 1 for the weights. Unsurprisingly, the supermodels trained with perfect observations are slightly better than the two supermodels trained with sparse and noisy observations. The differences however are very small. The synch rule supermodel trained with sparse and noisy observations performs least well, one would expect this result, since the weights for temperature are clearly a bit different from the other supermodels. Still, the model skill is not very far from the other supermodel skills.*

**Reviewer 2**

The supermodel approach is an ensemble approach in which models are combined dynamically. The idea is that when the weights that define the combinations are optimized by training on the basis of historical observations, the supermodel could potentially improve upon each of the models in predicting state trajectories. In other words, the supermodel is potentially a model that has better prediction performance than each of the model in the supermodel ensemble. In previous work the authors

**Table 1.** Weights for the CPT and synch rule supermodels trained with perfect observations, from Schevenhoven et al. (2019), and the weights for the CPT and synch rule supermodels trained with $\Delta T = 24$hr and 5% noise as denoted in Table 4 in this paper.

| model | T | VOR | DIV |
|---|---|---|---|
| CPT perf | 0.30 | 0.39 | 0.35 |
| synch perf | 0.35 | 0.38 | 0.34 |
| CPT noisy | 0.40 | 0.43 | 0.39 |
| synch noisy | 0.48 | 0.40 | 0.36 |

[Figure]

**Figure 1.** Forecast quality as measured by the RMSE of the truth and a model with a perturbed initial condition. The control is the difference between the perfect model and the perfect model with a perturbed initial condition.

demonstrated the viability of the supermodel approach in which weights were trained with the CPT (Cross pollination in time) algorithm and with the synch algorithm respectively. Simulations were performed in the context of the SPEEDO model with noiseless data observations at each time step. This paper builds upon that previous work. The extension in this paper is
125  that both the super model dynamics and algorithms are adapted to deal with noisy data that is sparse in time. Furthermore, a modification of the CPT algorithm is proposed to explore the possibility of negative supermodel weights, which were found previously found by the synch algorithm and showed to be beneficial. In this paper, experiments have been performed in the same SPEEDO context as in the earlier work, but now with noisy data that are only available every Delta T time steps. The weights that were found with the adapted CPT and synch rule were compared with the weights found in the previous work.
130  The result is that the adapted versions of the algorithm find similar weights as earlier, however now in the more realistic and difficult setting of the sparse and noisy data.

In my opinion the paper addresses relevant issues and contributes to the field. In particular the proposed methods could make a step forward towards the application of the supermodel approach in real world applications. In my opinion the paper is therefore publishable, however with revisions. Below I state some questions, comments and suggestions for revisions.

1. The organization/structure the paper should be improved.

1. a in my opinion, many parts of the 'Results' section rather belong to the 'Training methods' section. To be more specific, in section 3 'Training methods', the authors describe training algorithms, including the synch algorithm and CPT , as well as some novel adaptions in these algorithms. This is fine. However, in section 4, 'Results', the authors continue with describing several adaptations of the algorithm (synch rule adaptations, adaptations to nudging, CPT adaptation, CPT with negative weights). In my opinion, these are all training methods and belong to section 3 'Training methods' and not to section 4 'Results'. To summarize, I would suggest to put all the training methods in the section 'Training methods' and only results (findings from your experiments) in the 'Results' section.

*We would like to thank the Reviewer for this helpful suggestion. In this paper, we tried to distinguish the description of CPT and the synch rule as already published in earlier works from the new adaptations to the algorithms, with Sect. 3 containing the first part, and Sect. 4 "Results" containing the latter part. We agree that Sect. 3.2.1 "Nudging towards the observations" contains new findings, therefore we will integrate this section in Sect. 4.1.1 "Adaptations to nudging". We agree that Sect. 4.2 "CPT adaptations" fits better in Sect. 3 "Training methods", since it does not contain new findings. We will also change the title of the subsection to "Nudging". Furthermore, we will adjust the text in Sect. 3.3.2 "Duration of the training time", to remove elements that are reappointed in the results Sect. 4.*

1. b Section 3.1 has the title 'Training in SPEEDO'. It basically contains the SPEEDO imperfect model descriptions, which is important to appreciate the experimental set-up and the results. In my opinion section 3.1 is not about a training, and a renaming of section 3.1, e.g., 'SPEEDO imperfect models' would be appropriate in my opinion. I also would consider to move this subsection 3.1 to section 2 (Supermodels) , since this is more about the models that are to be combined in the SPEEDO supermodel than about Training methods.

*We would like to thank the Reviewer for the helpful suggestion. Section 3.1 could be divided into two parts: the first part about the SPEEDO imperfect models, and the second part specifically about training in SPEEDO, i.e. which physical variables are connected, how often, and where in space. We will move the first part to Sect. 2 "Supermodels".*

1. c There are more of these types of organizational issues. It would be good if the authors would have critical look at the paper and restructure it where necessary.

*We regarded the advice to have a critical look at the paper and restructure it where necessary. We will move the second part of Sect 3.1, "Training in Speedo", in which we describe which variables to exchange between the*

*models, and which frequency to use, to Sect. 4, in which we describe the specific Speedo experimental setup. This way, Sect. 3., "Training methods", focuses only on the CPT and synch rule training methods.*

2. Experimental set-up and interpretation of the outcomes

2. a Except for table 6, the findings (outcomes of the experiments – I am not talking about proposed training methods) in this paper are only supermodel weights (tables 3,4,5 +figure 3). It seems that the authors are only interested in the resulting weights. Why is that? Is model performance not more interesting? Although it seems nice that the weights found in the experiments are to some extend similar to the weights found in their previous paper, I have no idea about the effect of the difference in weighting on the performance of the super models. It would be good to have a table like table 6, with model performances for all the different combinations of (Delta T/noise/Training algorithm) in order to appreciate the benefit of the supermodel approach in the different cases. Ideally, these results are compared with performances from a weighted MME approach, but maybe that is too much to ask.

*We agree with the Reviewer that model performance is interesting. The focus of this paper is to make adaptations to the training methods in case the supermodels are trained based on the more realistic assumption of sparse and noise free observations. We know from Schevenhoven et al. (2019) the performance of supermodels with CPT and synch rule weights trained with perfect, noise free observations. The weights for the different experiments in Schevenhoven et al. (2019) were around 0.3± 0.05 for Model 1 and 0.7 ± 0.05 for Model 2. We did not find any significant difference in the forecast performance of the supermodels for these weights. Since the weights in the experiments in this paper are within this range, we think we can already estimate the outcome of model performance experiments. To see this point, we would like to refer to our answer given to point 9 of Reviewer 1, in which we compare the short term forecast performance of the supermodels trained by perfect observations, and the supermodels trained by observations available every 24 hours, with the highest noise level we used in this paper.*

*We show in Fig. 2 below, the Root-Mean-Squared-Error (RMSE) of supermodels based on noisy-sparse observations (s-CPT and s-synch noisy obs; with $\Delta T = 24hr$ and $\sigma = 5\%$) and of those based on perfect data (named - perf obs), for a 2 weeks forecast. See Table 2 for the weights. Unsurprisingly, the supermodels trained with perfect observations are slightly better than the two supermodels trained with sparse and noisy observations. The differences however are very small. The synch rule supermodel trained with sparse and noisy observations performs least well, one would expect this result, since the weights for temperature are clearly a bit different from the other supermodels. Still, the model skill is not very far from the other supermodel skills.*

*For a comparison with a weighted MME approach we would like to refer to Schevenhoven et al. (2019) Fig. 9 for a SPEEDO climatological wind comparison, and to Schevenhoven et al. (2019) Fig. 10 for a short term temperature forecast comparison.*

2. b Maybe I overlooked this, but how much training data was used for the synch rule?

*We apologize that this detail did not come apparent in our manuscript and thank the Reviewer to point it out. For*

**Table 2.** Weights for the CPT and synch rule supermodels trained with perfect observations, from Schevenhoven et al. (2019), and the weights for the CPT and synch rule supermodels trained with $\Delta T = 24$hr and 5% noise as denoted in Table 4 in this paper.

| model | T | VOR | DIV |
|---|---|---|---|
| CPT perf | 0.30 | 0.39 | 0.35 |
| synch perf | 0.35 | 0.38 | 0.34 |
| CPT noisy | 0.40 | 0.43 | 0.39 |
| synch noisy | 0.48 | 0.40 | 0.36 |

[Figure]

**Figure 2.** Forecast quality as measured by the RMSE of the truth and a model with a perturbed initial condition. The control is the difference between the perfect model and the perfect model with a perturbed initial condition.

*both the synch rule and CPT we use one year of training data. For the synch rule data is used continuously, while for CPT in bulks of 1 or 2 weeks. We will modify the text in Sect. 4.1 to make this clearer.*

*"[Following Schevenhoven et al. (2019), the training period is for both CPT and the synch rule equal to one year. For CPT every week, or every second week in case $\Delta T = 24$hr, the supermodel weights are calculated, and the model states are set back to the observations. For the synch rule the training period continues for an entire year. This amount of time is needed to obtain stable, converged weights.]"*

3. Some issues with the formal supermodel definition.

3. a It would be good to have a formal definition of the notion 'synchronized supermodel state'. From the paper I understand that a supermodel is synchronized if its models are in the same state, i.e., $x_1 = x_2$. I assume this in the comments below. I am not sure if this is correct.

*In our definition of a "synchronized supermodel state", we call it synchronization as long as the models inter-*

*act with each other such that there is a significant correlation in time between the different model trajectories.* *In the weighted supermodels as used in this paper, we impose a "perfect" synchronized state every time when $\delta_{mod(t,\Delta T)} = 0$. At this time we combine the model states by a weighted superposition into a supermodel state, that is given back to the individual models. Since the models share the same state space, the models continue their run with the exact supermodel state: $x_s = x_1 = x_2$. In previous works, instead of a weighted supermodel, we also used a so-called "connected" supermodel, in which the models are nudged towards each other. In a connected supermodel synchronization does not have to be perfect, dependent on the strength of the nudging coefficients there is more or less synchronization between the models. We will clarify our definition of synchronization in the first part of Sect. 2.*

*"[In our experiments so far, the models share the same state space, such that the models can continue with the exact supermodel state $\boldsymbol{x}_s$, implying perfect synchronization imposed between the models every $\Delta T$.]"*

3. b  I understand that in (1a, 1b, 1c) the supermodel remains synchronized if the states are initialized by $x_1 = x_2 = x_s$. However, in your description, it is not clear to me if this initialization is assumed or not. Although (1a,1b,1c) is not further used in the paper, it would be very helpful if the authors would state this explicitly.

*(1a, 1b, 1c) describes the original weighted supermodel formulation, in terms of combining model tendencies. In our SPEEDO context, we would indeed initialize the states by $x_1 = x_2 = x_s$: we will clarify this in the text. When using models with different structure, (a different resolution for example) this perfect initial synchronization might not be possible. Indeed, (1a, 1b, 1c) is not further used in the paper and, in (2a, 2b, 2c), we define a supermodel in which every $\Delta T$ the states are combined instead of the tendencies, which implies synchronization every $\Delta T$.*

*"[In Schevenhoven et al. (2019), we started for all models with the same initial conditions,]"*

3. c  About the redefined supermodel (2a,2b,2c), the authors state in line 98-99 that weighting the states ensures a synchronized supermodel state every Delta T. I understand that the state $x_s$ will be reset to $w_1 x_1 + w_2 x_2$ after every Delta T time steps, (eqn 2c) but I would expect that the states $x_1$ and $x_2$ should be synchronized to $x_s$ as well (or at least nudged to $x_s$) every Delta T. Otherwise, I would expect that $x_1$ and $x_2$ will diverge from each other. In particular if Delta T is large, the effect of the term $\delta_{mod(t,\Delta T)} f(x_s, p_1)$ on the dynamics of $x_1$ defined in (2a) will be negligible (the tendency of model 1 is only one time step governed by the superstate $x_s$ and all the other maybe 100s of time steps governed the state $x_1$.) In other words, the supermodel will basically behave as an unconnected ensemble of models, which I assume will not synchronize at all.

Please indicate whether or not $x_1$ and $x_2$ are synchronized (or nudged) to $x_s$ every Delta T steps and if this is not the case, please explain why my reasoning is erroneous.

*Every $\Delta T$ the supermodel state $x_s$ will be set to $w_1 x_1 + w_2 x_2$, and also the imperfect model states $x_1$ and $x_2$ will be replaced by the supermodel state $x_s$, indicated by the tendency term $\delta_{mod(t,\Delta T)} f(x_s, p_1)$. Hence, every $\Delta T$ the models are fully synchronized, by definition. Afterwards, indeed the models will diverge again, until brought back together after $\Delta T$. As discussed above in relation to point 8 of Reviewer 1 concerning the nonlinearity of*

*the models, $\Delta T$ should therefore not be too large. To give an indication how large $\Delta T$ can be: in synchronization experiments in the atmospheric component of SPEEDO, we found that combining the models every 24 hours still results in synchronized dynamical behavior.*

245 *"[In this study, we choose to let $\Delta T$ coincide with the observation frequency. The maximum time between two subsequent observations in this study is 24hr, this is frequent enough to maintain synchronization between the models.]"*

3. d The authors introduce a Delta T, on the one hand to reduce computational costs of combining models and on the other hand to deal with observational data that is not available at every time step (at least this is what I understand

250 from lines 85-90.) The authors seem to identify the Delta T in both cases. That is to say, the Delta T, which is the frequency of redefining the supermodel state in the supermodel (2a,2b,2c), introduced for computational efficiency, seems to be identified with the observation frequency Delta T as defined in e.g. line 312. Is it obvious that these should be the same? If there would bigger times between observations, would that mean that the dynamics of the free running supermodel should also have a larger Delta T, even if computational resources would allow for a much

255 smaller Delta T, or vice versa? What if training data is available at irregular times? Please explain or comment.

*In our study the observation frequency indeed coincides with the frequency of redefining the supermodel state, but this is not a necessary condition. We will clarify this in the text. The supermodel state should be defined frequent enough to maintain synchronization between the models, particularly when the observations are sparse. If there is not sufficiently available observational data the training might not be optimal, especially if one would work with*

260 *weights varying in time. If the training data is available at irregular times, one could try to find a greatest common divisor, use interpolation techniques, or in case there is a clear pattern in time, define the supermodel state during the forecast following this pattern.*

*"[ In this study, we choose to let $\Delta T$ coincide with the observation frequency. The maximum time between two subsequent observations in this study is 24hr, this is frequent enough to maintain synchronization between the*

265 *models. ]"*

4. Finally, two (minor) comments about statements in the Introduction

4. a Introduction, line 31-36: The authors remark that weights in a multimodal ensemble (MME) have the issue that weights that are optimal for model behavior in the past may not convert into optimal weights for the future. Then they remark that a dynamical on the fly method is desirable. By the statement 'Along this line, in the supermodel

270 approach ...' they seem to suggest that the supermodel approach is such an on-the-fly method that can deal with the issue. I don't see why this would be the case. Both the weights in a weighted MME and weights in a supermodel are necessarily to be trained on historical observations. If a certain type of change in model regime did not occur in the past but only in future, neither MME nor supermodel approach could have taken this into account and in both approaches, the weights might be suboptimal. Do the authors agree? Please comment.

275     *We would like to thank the Reviewer for this interesting question. The weights for both a weighted MME and a supermodel are indeed trained on historical observations, hence in supermodeling we might have the same issue that we do not have a qualitative good model for changes in model regimes that have not been seen in the observations during training. The MME mean approach is a statistical approach, while the intention of supermodeling is to be a dynamical one. Instead of acting afterwards on the resulting states, we act on the model equations. By acting*

280     *on the model tendencies during the run, "'on-the-fly", it might be possible to tackle the model errors at their root, and thereby changing and hopefully correcting the dynamics of the models. By correcting the dynamics, it might be possible that the supermodel displays improved behavior, even for regimes that have not been seen during the training.*

    *"[To cope with this, a "dynamical" on-the-fly approach to combine models is desirable, in which we act on the*

285     *model equations. Along this line, in the supermodel approach, models are combined during the simulation by sharing their own tendencies or states with each other, and not just their outputs as with the MME. This amounts to creating a new virtual model, the supermodel, that can potentially have better physical behaviour than the individual models. By combining the models dynamically into a supermodel, model errors can be reduced at an earlier stage, potentially mitigating error propagation and correcting the dynamics.]"*

290    4. b  Line 42-44: The authors make the comment that the if models in the supermodel are well enough synchronized, the supermodel can give improved model trajectories, whereas in MMEs individual variability in trajectories may be cancelled. I think the opposite might be true as well: by considering trajectories in an MME, one can compute a mean trajectory and fluctuations around this to assess the variability and the uncertainty of the prediction. Example: Model 1 predicts a severe storm tomorrow. Model 2 predicts the severe storm the day after tomorrow. It would be

295     very naïve to say that together they predict that there will be a mild storm tomorrow and the day after. To me a more natural interpretation would be: 50% chance that a severe storm arrives tomorrow and 50% chance that it arrives the day after tomorrow. Now back to the supermodel. The authors indeed mentioned that the models in a supermodel must be well enough synchronized. Does this mean that in a supermodel that is not well synchronized, there is risk that models are getting out of phase, counteract and damp each other (in case of the storm example,

300     that indeed the supermodel predicts a mild storm spread over two days?). Please comment.

    *We would like to thank the Reviewer for this very interesting discussion. We think indeed that if the supermodel is not well synchronized, there is a risk that the models are getting out of phase, counteract and damp each other. If there is a minimal amount of exchange between the models, we would expect to approximate the same result as when we would average trajectories that are not correlated, for example the trajectories from a multi model*

305     *ensemble. This probably would result in an unrealistic amount of variance reduction. In low dimensional systems, such as Lorenz63, we have seen a phenomenon we called "oscillation death": a system of loosely connected imperfect models potentially can have a significant lower variance than the imperfect models individually. Whether more complex models are also easily affected by oscillation death is still an open question We think there are cases*

310

*where only partial synchronization might benefit the supermodel performance. In Shen et al. (2016), two atmospheric models are connected to the same ocean model, with no interaction between the atmospheric components themselves, resulting in partial synchronization, mainly in the tropical Pacific. In this experiment the result is a better representation of the Intertropical Convergence Zone and the Pacific cold tongue in the supermodel than in the individual models.*

**References**

315    Carrassi, A., Bocquet, M., Bertino, L., and Evensen, G.: Data assimilation in the geosciences: An overview of methods, issues, and perspectives, WIREs Climate Change, 9, e535, https://doi.org/10.1002/wcc.535, 2018.

Du, H. and Smith, L. A.: Multi-model cross-pollination in time, Physica D: Nonlinear Phenomena, 353-354, 31 – 38, https://doi.org/https://doi.org/10.1016/j.physd.2017.06.001, 2017.

Hagedorn, R., Doblas-Reyes, F. J., and Palmer, T.: The rationale behind the success of multi-model ensembles in seasonal forecasting — I.
320    Basic concept, Tellus A: Dynamic Meteorology and Oceanography, 57, 219–233, https://doi.org/10.3402/tellusa.v57i3.14657, 2005.

Schevenhoven, F., Selten, F., Carrassi, A., and Keenlyside, N.: Improving weather and climate predictions by training of supermodels, Earth System Dynamics, 10, 789–807, https://doi.org/10.5194/esd-10-789-2019, 2019.

Schevenhoven, F. J. and Selten, F. M.: An efficient training scheme for supermodels, Earth System Dynamics, 8, 429–438, https://doi.org/10.5194/esd-8-429-2017, 2017.

325    Shen, M.-L., Keenlyside, N., Selten, F., Wiegerinck, W., and Duane, G. S.: Dynamically combining climate models to "supermodel" the tropical Pacific, Geophysical Research Letters, 43, 359–366, https://doi.org/10.1002/2015GL066562, 2015GL066562, 2016.

---

## Author Response (AR2)

**Response to Editor**

Francine Schevenhoven[1,2,3] and Alberto Carrassi[4,5]

[1]Geophysical Institute, University of Bergen, Bergen, Norway
[2]Bjerknes Centre for Climate Research, Bergen, Norway
[3]University of Colorado, Boulder, USA
[4]Dept. of Meteorology and NCEO, University of Reading, United Kingdom
[5]Department of Physics and Astronomy "Augusto Righi", University of Bologna, Bologna, Italy

We are very thankful to the Editor for her helpful comments. We revised the manuscript accordingly and provide a reply below. We hope that the current version of the manuscript fits with the GMD standard.

*"The exact version of the SPEEDO model code with the CPT and synch rule training integrated that is used to produce*

5    *the results used in this paper is archived on Zenodo (Schevenhoven, 2021), as are input data and scripts to run the model for all the simulations presented in this paper"*

*The above code and data availability statement does not enable me to identify the relevant parts of the the Zenodo archive. I can see 3 downloads on Zenodo, but it is not clear what they are. My preference would be for a more detailed introduction on the Zenodo page, but alternatively (or additionally) you could extend the availability section in the manuscript to include the*

10    *names of the relevant parts of the archive (eg model code, training code, data, scripts). Are the model outputs and scripts used to create the figures also included in the archive?*

We apologise for not having given a clearer description of the files in the Zenodo archive (Schevenhoven, 2021). The files that were already uploaded on Zenodo consisted of the atmosphere (AtmosCoupler), ocean (CLIO) and land (LBM) component of SPEEDO, where the atmosphere part also contained the CPT and synch rule training. We added a folder postpro-

15    cess_SPEEDO, with the relevant files to postprocess the data and scripts to produce the figures in this manuscript, and a folder Results_SPEEDO, containing the model output for the different experiments. Furthermore, we added a folder called progsandlibs to the Zenodo archive, that contains the necessary programs and libraries to install SPEEDO with multiple communicating atmospheres through MPI. To clarify the content of the files, we extended the description on Zenodo, with a short overview per directory. Furthermore, in the code availability section of the manuscript we provided a concise description of the Zenodo

20    files.

**References**

Schevenhoven, F.: Supermodel training: CPT and the synch rule on SPEEDO - v.1, https://doi.org/10.5281/zenodo.6244858, 2021.